# POISONING GENERATIVE MODELS TO PROMOTE CATASTROPHIC FORGETTING

## ABSTRACT

Generative models have grown into the workhorse of many state-of-the-art machine learning methods. However, their vulnerability under poisoning attacks has been largely understudied. In this work, we investigate this issue in the context of continual learning, where generative replayers are utilized to tackle catastrophic forgetting. By developing a novel customization of dirty-label input-aware backdoors to the online setting, our attacker manages to stealthily promote forgetting while retaining high accuracy at the current task and sustaining strong defenders. Our approach taps into an intriguing property of generative models, namely that they cannot well capture input-dependent triggers. Experiments on four standard datasets corroborate the poisoner's effectiveness.

## 1 INTRODUCTION

The vulnerability of machine learning systems must be scrutinized before they can be deployed to security-critical applications. The common evasion attack assumes that clean target instances can be manipulated at test time, which can be unrealistic in many scenarios. In contrast, poisoning attacks only make malicious and imperceptible modifications to the training set, so that the prediction on test examples can be mistaken. The threat models may insert poison examples (Chen et al., 2017), flip the training labels (Xiao et al., 2012; Levine & Feizi, 2021), or modify the training example inputs (Biggio et al., 2012; Shafahi et al., 2018).

Although poisoning attacks have been extensively studied under discriminative learning, their potential risk in *generative* learning has been largely understudied. Ding et al. (2019) poisons the training examples so that the learned generator covertly changes some important part of the output image, e.g., turning a red light into green. Salem et al. (2020) enables the adversary to control the output image by planting a trigger in the input image or noise. Both are backdoor attacks requiring write access to test data, and work in batched learning scenarios.

The increasing penetration of generative models in machine learning urges the investigation of poisoning attacks in a broader range of learning paradigms In this work, we focus on continual learning, a prominent setting where tasks arrive in streams and each of them corresponds to a discriminative learning problem such as classification (Chen & Liu, 2018). Since the tasks are streamed and cannot be stored, the running classifier often suffers from *catastrophic forgetting*, where the performance on older tasks gradually deteriorates (McCloskey & Cohen, 1989).

Deep generative replay (DGR) is a natural tool to bring back the memory of the previous tasks by learning a generative model to fit the data of these tasks (Shin et al., 2017; Cong et al., 2020). Despite their effectiveness, new vulnerabilities are also opened up where misleading examples can be injected to the training data $\mathcal{D}_t$ for the current task $t$, so that catastrophic forgetting can be *promoted* when such poisoned $\mathcal{D}_t$ is used for training both the replayer $G_t$ and the classifier. In this work, we seek practical and stealthy poisoning attacks on DGR that achieve three objectives:

**O1** After moving past task $t$, the classifier will soon forget what was learned from it (i.e., perform poorly on clean test examples drawn from it) despite using a replayer for all the tasks seen so far.

**O2** *During* task $t$, the classifier trained from the poisoned data does *not* suffer degradation of test accuracy on task $t$ itself. This is important because poor performance on the current task can raise significant and immediate suspicion. In contrast, by promoting forgetting, the harm will

manifest itself only after the victim has moved on to the next task, by which time it will have become too late because the access to the samples of task $t$ is already lost.

**O3** The poisoned data should be robust to solid defenses deployed by both the classifier and the replayer.

The main difficulty lies in two folds. Firstly, although both **O1** and **O2** are straightforward to fulfill individually, they are at odds with each other and are hard to fulfill simultaneously. Secondly, the transiency of data streams compels the adversary to make irrevocable attacks at task $t$ before future tasks arrive and before catastrophic forgetting can start to occur. Due to this difficulty, poisoning attacks have been much less studied in an online setting. Mladenovic et al. (2022) addressed the online decision problem of selecting $k$ examples for evasion attack. Zhang et al. (2020) assumed the instances are drawn i.i.d. from a time-invariant distribution, which is not the case in continual learning because tasks may even have disjoint classes. Other works require multiple passes of the data stream (Gong et al., 2019; Lin et al., 2017; Sun et al., 2020), or clairvoyant knowledge of future data (Burkard & Lagesse, 2017; Wang & Chaudhuri, 2018).

**Our contribution**, therefore, is to overcome these challenges and to reveal the vulnerability of generative models in the sense that their training data can be poisoned stealthily such that a task can be learned well at present but forgotten soon in the future. Noting that simple label-flipping poisoning can be easily detected, we resort to dirty-label backdoor/Trojan attack (Liu et al., 2018) to attain **O2**: the trained classifier performs correctly on clean examples, but errs if the example is planted with a trigger. To further achieve **O1** and **O3**, we capitalize on the input-aware backdoor (Nguyen & Tran, 2020), which allows the trigger to vary depending on the image. As a result, it can not only withstand stronger defense (§4), but also enjoys higher variation and stealthiness, hence much harder for a generative model to capture. So the replayed images do not well preserve the trigger (we call it trigger-discarding property in §3.3) while retaining the incorrect label, leading naturally to forgetting (§3). The problem is set up in §2, and experiments are provided in §5 to show the effectiveness of the attack. Our innovations are summarized as follows:

- Proposing the first poisoning attack that promotes catastrophic forgetting in continual learning.
- Achieving poisoning (no trigger is needed at test time) through a novel way of leveraging backdoor attack that is particularly effective for exacerbating catastrophic forgetting.
- Identifying a trigger-discarding property of generative models that is intriguing for backdoor attack.

**Related work** Generative models have been pervasive in machine learning (Murphy, 2023, Part IV), reaching far beyond the original role of density estimation and serving as a key infrastructure in supervised, unsupervised, and reinforcement learning. We contend that their vulnerability needs to be examined in the context of their use. In the vanilla density estimation, Condessa & Kolter (2020) learned robust variational auto-encoder (VAEs) that retain high likelihood for the data points under adversarial perturbation. The underlying threat is evasion attack, and along similar lines, Tabacof et al. (2016) and Kos et al. (2018) studied attacks that promote reconstruction error of the decoder in a VAE. Some recent works address attacks on membership inference (Hayes et al., 2019; Chen et al., 2020; Hilprecht et al., 2019), model extraction (Hu & Pang, 2021), and attribute inference (Stadler et al., 2022). However, poisoning attacks on generative models are still understudied.

Our aim is to poison a generative model instead of learning a generative model to produce poisons for another (discriminative) model (Yang et al., 2017; Muñoz-González et al., 2019). We also leave it as future work to defend the proposed attack, noting that (certifiable) defense and detection have been well studied for poisoning attack on *batch* discriminative models (Peri et al., 2019; Steinhardt et al., 2017; Levine & Feizi, 2021; Jagielski et al., 2018).

## 2 ATTACKING GENERATIVE MODELS IN CONTINUAL LEARNING

We consider the continual learning setting, where tasks arrive sequentially. The goal is to keep updating a classifier that predicts accurately not only on the current task, but also on the previous tasks. Each task $t$ is indeed a joint distribution $P_t(X, Y)$, where $X \in \mathcal{X}$ is the input from a feature space $\mathcal{X}$, and $Y \in \mathcal{Y}_t$ is the label whose domain $\mathcal{Y}_t$ may change with the task. For example, $\mathcal{Y}_1$ consists of digits 0 and 1, and $\mathcal{Y}_2$ encompasses 2 and 3. Even in the case where the domains remain constant, the distribution $P_t$ can shift. The goal of continual learning is to find a classifier $C_t$, such that the overall risk across all tasks seen *so far* is minimized:

$$C_t \approx \arg\min_C \sum_{i=1}^{t} \mathbb{E}_{(X,Y)\sim P_i}[\ell(C(X), Y)]. \tag{1}$$

Here $\ell$ is a loss function, and we will focus on multi-class classification with cross-entropy loss. A major challenge in continual learning is that at task $t$, only the samples from the current $P_t$ are available, while those from the previous tasks are no longer accessible. Although most algorithms would only adapt to the new task instead of completely retraining from scratch, the performance on previous tasks may still deteriorate significantly, a phenomenon known as *catastrophic forgetting*.

To alleviate this issue, DGR-based approaches resort to learning a DGR model $G_t$ that approximately replicates $P_t$. Due to the constraints in computation and storage, it has to be a lossy approximation because otherwise one might as well store all the past examples. Then at each task $t$, samples of $(X, Y)$ pairs are drawn from not only the current $P_t$, but also from the replayers for the previous tasks $G_{1:t-1} := \{G_i\}_{i=1}^{t-1}$. Their union is subsequently used to update the classifier into $C_t$. The whole process of vanilla DGR-based continual learning is illustrated in Algorithm 1.

DRG models can be simplified into a **single** replayer $G$ that is updated over time, as opposed to one replayer per task. However, our contribution is the **poisoner**, not the replayer. Employing multiple replayers only makes attacks even more challenging, because instead of just poisoning one running replayer, We are now tasked to poison many of them, each of which can only be poisoned once at its current task. If we simply keep a single running replayer, then we enjoy many opportunities of poisoning it at any time and our objectives will become much easier to achieve. To conclude, multiple replayers set up a more stringent benchmark for testing our poisoner. A natural choice of the replayer is a conditional generative model such as conditional GAN (cGAN), which first samples the label $Y$ from a discrete distribution, and then generates the feature $X$ via the cGAN.

## 2.1 ATTACKERS AND LEARNERS

Two parties participate in the process, and we first set forth our assumptions on them. The **victim learner/user** consists of a *classifier*, a *replayer*, and a *defender*. We assume **none of them has access to the original clean data**, and can only access the poisoned data. Further, the replayer must perform well, i.e., the generated samples match the distribution of data presented to it for training. Otherwise, the replayer would not be adopted by the user in the first place, and can spare any need of attack by, e.g., generating random images with random labels. We also assume that the learner cannot store any data beyond its current task, which is standard in continual learning.

**Algorithm 1:** Deep generative replay (DGR) used by continual learning to combat catastrophic forgetting

---

**Input:** Tasks $1, 2, \ldots$ represented as $P_1, P_2, \ldots$
1 Initialize classifier $C_0$
2 **for** $t = 1, 2, \ldots$ **do**
3     **for** $i \in [t-1] := \{1, 2, \ldots, t-1\}$ **do**
4         $\mathcal{S}_i \leftarrow$ **SampleFromDGR**$(G_i)$ $(X, Y$ pairs$)$
5     Sample $\mathcal{D}_t$ from task $P_t$     $\leftarrow$ to be poisoned
6     $C_t \leftarrow$ **TrainClassifier**$(C_{t-1}, \mathcal{D}_t \cup \mathcal{S}_{1:t-1})$
7     $G_t \leftarrow$ **TrainReplayer**$(\mathcal{D}_t)$, e.g. conditional GAN

A defender is an algorithm that the learner employs to scrutinize and prune the possible poisons in the training data. In DGR, it means examining the data collected from the current task $t$, as well as the replayed samples from previous $G_i$ in step 4. This is known as *pre-training* defense, whose counterpart—*post-training* defense—patches up the learned model (Wang et al., 2019).

The **attacker** (our threat model) is *only allowed to poison (modify) the samples $\mathcal{D}_t$ in step 5 of Algorithm 1*. The attacker has no access to the internal mechanism of classifier, replayer, or defender. It can *read* the gradient of the classifier's training objective with respect to the input, but not its structure or weights. This is a moderate mid-ground between full access (e.g., training by the attacker itself on the cloud) and no access (independently manipulating the training examples). Such an assumption has been commonly adopted, e.g., by dynamic backdoors (Nguyen & Tran, 2020), Witches' brew (Geiping et al., 2021), and by implicit differentiation based poisoning (Muñoz-González et al., 2017). We will pursue a dirty-label backdoor attack, i.e., a small portion $\rho_b$ of $\mathcal{D}_t$ will flip their label, along with a trigger of size $\rho_a$ planted to their input image (more details in §3.2).

**Achieving poisoning through backdoor.** Although our method will leverage backdoor attacks, our overall goal is poisoning attack, *not* backdoor. In backdoor attacks, a classifier predicts poorly only on backdoored examples with triggers, while remaining well on clean test examples. In contrast, poisoning attacks (excluding backdoor) aim to predict poorly on *all* examples, irrelevant of "trigger".

In continual learning, an attacker generally does not have the liberty of planting a trigger on test examples. So we will address in **O1** and **O2** a much more challenging setting (from an attacker's perspective) where such an access is not available. It is important to note that our approach only utilizes backdoor attacks as a means of achieving the goal of poisoning the generator/replayer.

## 2.2 DIFFICULTIES IN THE ATTACK

Since most of the existing poisoning algorithms are in the batch setting, the extension to continual learning brings about new and significant challenges.

Firstly, the poison cannot compromise the *current* classifier, but should sufficiently poison the DGR so that the samples drawn from it during the *later* tasks will be detrimental enough to forget the previously learned tasks. This rules out simply flipping the label of some examples in $\mathcal{D}_t$, because it is easy to detect (Levine & Feizi, 2021) and the resulting classifier $C_t$ will perform poorly on task $t$.

Secondly, since the future tasks have not been witnessed yet at task $t$ and the training of $C_{t+1}$ has not started, it is infeasible to optimize the forget-inducing distortion on $\mathcal{D}_t$ via back-propagation based optimization – the context and objective are not yet available for future forgetting.

## 3 THE ATTACK ALGORITHM

Our poisoning attack proposes evading the defense by leveraging the *input-aware backdoor* attack (Nguyen & Tran, 2020), so that *mislabeled* data points carrying a *trigger* can be injected to $\mathcal{D}_t$ in a small amount. In particular, our approach achieves **O1** to **O3** through the following effects:

1. Since the triggers depend on (hence vary across) the input data, the defender can hardly detect it.
2. Since the mislabeled examples for the current task all carry an input-aware trigger, the learned backdoored classifier for task $t$ makes mistakes only for backdoored examples. As such, it predicts accurately on pristine test examples for task $t$ which carry no trigger.
3. As generative models essentially represent a lossy compression, it is generally unable to capture the triggers that change with the input. When replayed later for a future classifier at task $t+1, t+2, \ldots$, the triggers go absent while the incorrect label is retained. So the classifier will be trained on mislabeled examples of task $t$ with no backdoor, hence misclassifying clean test examples.

### 3.1 BACKDOOR ATTACKS

Our solution is based on backdoor attacks, which despite the marked resemblance to poisoning attacks, do *not* misclassify a test example unless a pre-designed trigger is inserted to it. Such a flexibility of modulation proves essential. Backdoor attacks such as BadNets (Gu et al., 2019) plant a pre-selected or learned trigger into some training images at a pre-selected or varying location. A number of variations are available such as soft blending and multi-channel. Adding the resulting image to the training data along with a flipped label (randomly selected for an untargeted attack, and pre-specified for a targeted attack), a classifier can be trained that enjoys two important properties:

**P1** Once a test image is also backdoored with a trigger, the predicted label will change to the pre-specified (or random) one in a targeted (or untargetted) attack.

**P2** However, the test accuracy on pristine images (without a trigger) can remain very high.

**Our inspiration** originates from this trigger-based modulation. Suppose we plant the trigger on $\mathcal{D}_t$ in step 5 of Algorithm 1, and denote the resulting training set as $\tilde{\mathcal{D}}_t$. Then the resulting $C_t$ will perform well on pristine test examples of task $t$, because they do not carry the trigger. After moving to task $t + 1$, the replayer will (approximately) reproduce $\tilde{\mathcal{D}}_t$, at which point two cases can be considered:

- If the replayer works purely by rote, then $\tilde{\mathcal{D}}_t$ will be exactly replayed and the resulting classifier $C_{t+1}$ will be backdoored in the same way as $C_t$. As a result, it will still predict accurately on clean samples from task $t$, i.e., the attack fails in promoting forgetting.
- If the replayer is lossy and is unable to capture or reproduce the trigger, then the replayed examples *might* no longer carry the trigger. However, they still carry the flipped label. As a result, $C_{t+1}$ will now be trained on mislabeled examples without a trigger, and will therefore perform poorly on task $t$. In this case, the attacker successfully promoted forgetting.

The requirement on the replayer in the second case may appear unrealistic, because firstly the replayer is supposed to faithfully preserve the salient information in the inputs to address catastrophic forgetting. Secondly, a user (who constructs and trains the replayer) obviously has no motivation to collaborate with an attacker. Therefore, the key challenge for the attacker is to design delicate triggers that are *as likely to be overlooked and disregarded as possible* under generative modeling, while retaining the good performance on clean test data during task $t$ (property **P2**).

---

**Algorithm 2: InputAwareBackdoor**

---

**Input:** Data generation distribution $P(X, Y)$, which will the invoked with $P = P_t$ at task $t$

1 $(\tilde{\mathcal{D}}, \mathcal{L}_{div}) \leftarrow$ **InputAwareBackdoor–Obj**$(P, B_{[\mathcal{Y}]})$

2 **return** $\arg\min_{C, B_{[\mathcal{Y}]}} \{\mathcal{L}_{cl} + \lambda_{div}\mathcal{L}_{div}\}$, *where $\mathcal{L}_{cl} = \sum_{(x,y)\in\tilde{\mathcal{D}}} \ell(C(x), y)$ is the classif. risk*

---

**Algorithm 3: InputAwareBackdoor–Obj**

---

**Input:** $P$ as in **InputAwareBackdoor**, and backdoor generators $B_{[\mathcal{Y}]} := \{B_y : y \in \mathcal{Y}\}$

1 Initialize $\tilde{\mathcal{D}} = \emptyset$ which will contain clean and poisoned examples. Set $\mathcal{L}_{div} = 0$ (diversity loss).

2 **for** $(x, y)$ *sampled from $P$ for task $t$* **do**

3      Sample $d \sim U(0, 1)$,    sample $(\hat{x}, \hat{y})$ from $P$ excluding $(x, y)$,    sample $y'$ from $\mathcal{Y}\backslash\{y\}$,

4      $\mathcal{L}_{div}$ += $\|x - \hat{x}\| / \|B_{y'}(x) - B_{y'}(\hat{x})\|$  // Accrue the diversity loss

5      **if** $d < \rho_b$ **then**   $x' \leftarrow x \odot B_{y'}(x)$,   $\tilde{\mathcal{D}}$ += $(x', y')$  // make a backdoor example

6      **else if** $d < \rho_b + \rho_c$ **then** $x' \leftarrow x \odot B_{y'}(\hat{x})$,   $\tilde{\mathcal{D}}$ += $(x', y)$  // make a cross example

7      **else**   $\tilde{\mathcal{D}}$ += $(x, y)$  // clean example

8 **return** $\tilde{\mathcal{D}}$ and $\mathcal{L}_{div}$

---

This is indeed challenging as we experimented. Static backdoor (BadNet) can be easily replayed by a generative model. Trojan attack requires access to the victim model's structure and weights. Neither can it survive the defense of neural cleansing. We also tested static backdoor with changing location, which again, turned out easily detected and fixed by neural cleansing. Witches' Brew (Geiping et al., 2021) and other gradient matching based methods need to know the target before deploying the attack, while future tasks are unknown in continual learning. Eventually, it turns out the *input-aware* backdoor satisfies our need, where a trigger is customized for each example through a learnable generative model, hence exhibiting much less regularity for the replayer to capture.

### 3.2 INPUT-AWARE BACKDOOR

We first recap the input-aware backdoor (IAB, Nguyen & Tran, 2020) as shown in Algorithm 2 under a given data distribution $P$, and then detail how to utilize it for our purpose. In line 2, the classifier $C$ and class-wise backdoor generating *networks* $B_y$ are jointly optimized over an objective constructed in Algorithm 3, where each $(x, y)$ sampled from $P$ contributes in one of the following modes:

- As a backdoor example with probability $\rho_b$: a wrong label $y'$ is randomly picked, and then a trigger that depends on $x$ is generated by $B_{y'}(x)$ and injected to $x$ in line 5 via elementwise product $\odot$.

- As a cross-trigger example with probability $\rho_c$. To ensure that a trigger synthesized for one example is *not* effective for another, another $\hat{x}$ is sampled from $P$ with label $\hat{y}$. Then $x$ is injected with the trigger generated from $\hat{x}$ for a wrong label $y'$, and the result is paired with the clean label $y$ (line 6).

- As a clean example otherwise (line 7).

In Algorithm 2, $B_{[\mathcal{Y}]}$ is explicitized in line 1 to stress that both the diversity loss $\mathcal{L}_{div}$ and classification risk $\mathcal{L}_{cl}$ (through $\tilde{\mathcal{D}}$) are functions of $B_{[\mathcal{Y}]}$, which is then optimized in line 2 along with the classifier $C$. To see that the attacker fits in our threat model, note $C$ and $B$ are jointly optimized in line 2 of Algorithm 2, and the attacker only requires *reading* the gradient with respect to the input of $C$. As observed in our experiment, IAB proffers the following property (Nguyen & Tran, 2020):

**P3** The backdoor in IAB can be hardly detected by state-of-the-art methods, e.g., neural cleansing.

To summarize, we fulfilled **O2** by **P2**, and **O3** by **P3**. To meet **O1**, we require a *trigger-discarding* property as follows, which plays a key role in our method and will be discussed in the next subsection:

**P4** The replayer cannot well capture the trigger generation network of IAB, in the sense that the replayed examples do *not* well preserve the triggers.

### 3.3 TRIGGER-DISCARDING GENERATIVE MODELS

Property **P4** depends on both the replayer and the trigger. If the trigger is a constant small white square at the image center, most generative models will preserve it. Same is true if the replayer only replicates the training set. In general, it is supposed to capture the salient features of the input, and one might presume that triggers are likely to be discarded if they vary a lot across examples. It turns

out not true. For example, we placed a square/triangle/round in random colors at random positions of the images, and a WGAN easily reproduced them with these (interpolated) color, shape, and position.

Formally, let $P_x$ be the data distribution, and suppose given an input $x$, the trigger is generated by a learnable network $f_\theta(x)$ and is added to $x$ by a pre-specified operation $g(x, f_\theta(x))$. It induces a distribution of backdoored examples as a push-forward of $P_x$: $Q_x^\theta := (x \mapsto g(x, f_\theta(x))) \# P_x$. Let a generative learning algorithm $\mathcal{A}$ map a set of examples $\{x_i\}_i$ to a distribution. Then the trigger is intended to *demote* some divergence (e.g., KL and Wasserstein) between

$$P_x \quad \text{and} \quad \mathbb{E}_{x_i \sim Q_x^\theta} \mathcal{A}(\{x_i\}_i). \tag{2}$$

Directly computing the gradient in $\theta$ is both expensive and infeasible, because $\mathcal{A}$ is assumed inaccessible in §2.1. Fortunately, our experiments show that **P4** is well (but not perfectly) achieved when IAB is applied in conjunction with several SOTA generative models such as conditional Wasserstein GAN (cWGAN, Engelmann & Lessmann, 2021) and conditional VAE (Sohn et al., 2015). This is no surprise because these models have limited capacity, and the trigger's dependency on the input, which is more involved than just random, significantly raises the sample complexity for generative learning. A theoretical analysis is left for future work, and §5.2 empirically illustrates this intriguing property.

### 3.4 Poisoning the replayer via IAB

We are now ready to apply IAB to poison the DGR used by continual learning against forgetting. We will call our method Continual Input-Aware Poisoning (CIAP). It does *not* backdoor test images.

Algorithm 4 demonstrates the operation of all participants (user, attacker, and defender) during task $t$, which corresponds to the loop under a given $t$ in Algorithm 1 (line 3 to 7 therein). The red colored steps are reserved for defense, which will be detailed in Section 4. In line 7, the poisoned data $\tilde{\mathcal{D}}_t$ for task $t$ is joined with the replayed data $S_{1:t-1}$ to construct the classification risk $\mathcal{L}_{cl}$. Both the attacker and the user are trained in the same way as in Algorithm 2, while the only difference is that our optimization here is based on mini-batches, and the replayer $G_t$ is additionally learned in line 10.

### 4 The Defender

We consider two defenses against the CIAP attack. The first is neural cleansing (Wang et al., 2019), which has been inserted in line 11 of Algorithm 4. If it were successful, then the backdoor planted in $C_t$ would be detected and removed, thereby defeating objective **O2** with immediate poor accuracy on clean test examples at task $t$.

---

**Algorithm 4:** Operation of the user, attacker, and defender during task $t$ (in place of line 3 to 7 of Algorithm 1)

**Input:** $P_t$ for task $t$
**Input:** Classifier $C_{t-1}$, and backdoor generators $B_{[\mathcal{Y}]}$
1 Initialize $C$ with $C_{t-1}$.
2 **for** $i \in [t-1]$ **do**
3     $\mathcal{S}_i \leftarrow$ **SampleFromReplayer**$(G_i)$ $(X, Y$ pairs$)$
4 **Defender**: Apply $\nu$-SVM on the replayed data $\mathcal{S}_{1:t-1}$
5 **for** *number of iteration (**run in mini-batches**)* **do**
6     $\tilde{\mathcal{D}}_t, \mathcal{L}_{div} \leftarrow$ **InputAwareBackdoor–Obj**$(P_t, B_{[\mathcal{Y}]})$
       // Both $\tilde{\mathcal{D}}_t$ and $\mathcal{L}_{div}$ are functions of $B_{[\mathcal{Y}]}$
7     $\mathcal{L}_{cl} \leftarrow$ sum of $\ell(C(x), y)$ over $(x, y) \in \tilde{\mathcal{D}}_t \cup \mathcal{S}_{1:t-1}$
       // $\mathcal{L}_{cl}$ is a function of $C$ and $B_{[\mathcal{Y}]}$
8     **User**: update $C$ to reduce $\mathcal{L}_{cl}$
9     **Attacker**: update $B_{[\mathcal{Y}]}$ to reduce $\mathcal{L}_{cl} + \lambda_{div}\mathcal{L}_{div}$
10     Update the replayer $G_t$ using $\tilde{\mathcal{D}}_t$ and the latest $B_{[\mathcal{Y}]}$
11 **Defender**: Neural Cleansing on $C_t \leftarrow C$
12 **return** $G_t, B_{[\mathcal{Y}]}$, *and* $C_t$

---

Our second defense is aimed at objective **O1**. To this end, we apply an outlier detector $\nu$-SVM to $\mathcal{S}_{1:t-1}$, which is in line 4 of Algorithm 4. If it managed to filter out mislabeled replayed samples, then the attacker would fail to bolster catastrophic forgetting. Here $\nu$ is a hyperparameter controlling the fraction of outliers. Since its value is unknown in practice, our experiment will enumerate a range of $\nu$ values, and demonstrate the extent to which the learner's performance can be saved respectively.

**It is crucial to recognize that the replayed examples are *not* simply label-flipped poisons** (i.e., clean images with a wrong label), although the replayer is poisoned with label-flipped and backdoored examples. This is for two reasons. Firstly, since the examples of a class $y'$ is fed to the replayer to train for the class $y$, the generation of the features/images for class $y$ is contaminated. Secondly, the input-dependent triggers introduce additional complications to the generative model. Indeed, we tested by directly generating label-flipped examples based on clean images, and $\nu$-SVM easily filtered them out. However, this is not the case when $\nu$-SVM is applied to our replayed images (§5.3).

# 5 EXPERIMENTAL RESULTS

We next experiment on CIAP to verify: i) it attains the two objectives **O1** and **O2**; ii) the trigger-discarding property introduced in §3.3 holds true for commonly used generative models; iii) CIAP remains effective under strong defenders (**O3**). The code is available at Online Supplementary.

## 5.1 EFFECTIVENESS OF THE ATTACK FOR OBJECTIVES **O1** AND **O2**

We tested CIAP on five datasets: split-MNIST (Ciresan et al., 2011), split-CIFAR-10 (Krizhevsky & Hinton, 2009), FashionMNIST-MNIST (Xiao et al., 2017), permuted-MNIST (Goodfellow et al., 2014), and split-EMNIST (Cohen et al., 2017). We used SpinalVGG as the victim classifier (Kabir et al., 2020) for the four MNIST datasets, and ResNet (He et al., 2016) for the split-CIFAR-10 dataset. The results shown here use cWGAN with gradient penalty as the replayer, and more results for cVAE are deferred to Appendix E. The poison ratio $\rho_b = 0.25$, and the cross ratio $\rho_c = 0.15$. Each trigger was allowed to change $10\%$ of the pixels of a selected image (mask density), and Figure 2 will show that the triggers are quite inconspicuous.

**split-MNIST** We separated the entire dataset of MNIST into five tasks, each consisting of images from two disjoint classes in MNIST – the first task includes classes 0 and 1; the second task includes 2 and 3; and so on. The victim model was trained for 100 epoch in each task.

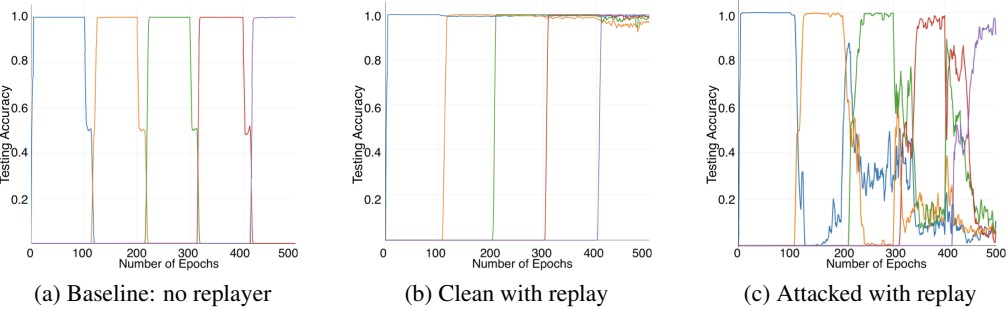

(a) Baseline: no replayer     (b) Clean with replay     (c) Attacked with replay

Figure 1: Test accuracy on clean *test* images for **split-MNIST**

Figure 1a shows the baseline result without a replayer, where the blue line represents the test accuracy of the first task, orange line for the second, etc. As expected, the test accuracy for each task drops rapidly to 0% after the victim model proceeds to a new task. It is 0% because the new task has no overlap with the previous ones in the label space. Figure 1b shows the result of DGR-facilitated training, where the forgetting is significantly mitigated, and the test accuracy remains high on all trained tasks. This confirms the effectiveness of DGR and the sufficient capacity of the cWGAN.

Figure 1c shows the result after our attack CIAP is enacted. The test accuracy of each current task can still achieve nearly 100%, corroborating the achievement of objective **O2**. When the learner moves to the next task, the accuracy on the previous tasks falls significantly to around 20% despite some fluctuations. This confirms that the objective **O1** (forgetting) has also been attained.

Finally, we plot in Figure 2 some example clean images of class 0 and 1 from the first

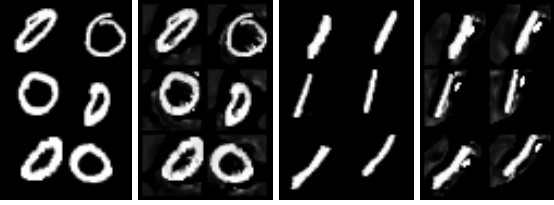

(a) Clean 0   (b) Poisoned 0   (c) Clean 1   (d) Poisoned 1

Figure 2: The clean and poisoned images used to train the replayer for **split-MNIST**.

task of split-MNIST, along with their corresponding poisoned images constructed by IAB (before label flipping). Clearly the poisons are quite inconspicuous.

**split-CIFAR-10** To illustrate the effectiveness of CIAP in colored space, we repeated the experiment on CIFAR-10, with the same setup of five disjoint tasks. Here the victim model was trained for 50 epochs on each task.

Similar to the split-MNIST, the victim model completely forgets the earlier trained tasks after a new task starts, as shown in Figure 3a. After DGR is introduced in Figure 3b, although forgetting is not as

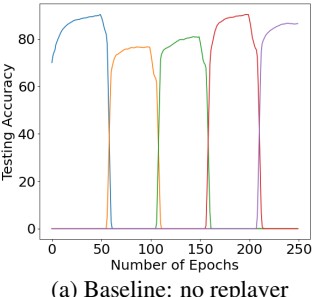 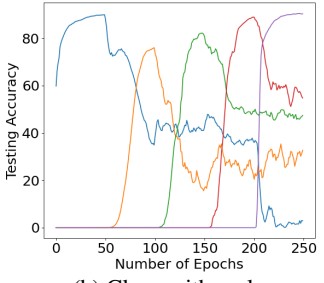 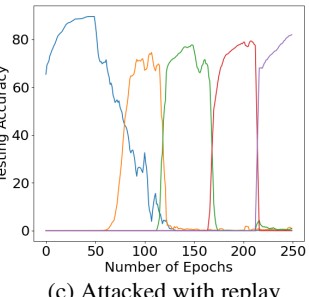

(a) Baseline: no replayer      (b) Clean with replay      (c) Attacked with replay

Figure 3: Test accuracy on clean test images for **split-CIFAR-10**

well mitigated as in split-MNIST, solid improvement is still made on the test accuracy for all trained tasks. However, the improvements brought by DGR were completely eliminated by the CIAP attack. As Figure 3c shows, the test accuracy on past tasks drops down to 0% after being poisoned. During the current task, however, the accuracy can still achieve the same level as in Figure 3a.

We also tested on the split-EMNIST dataset with 10 tasks, and the results are similar; see Appendix C. The results of FashionMNIST-MNIST and permuted-MNIST are in Appendix A and B, respectively.

## 5.2 Investigation of the trigger-discarding property

To better illustrate property **P4**, we set up two experiments on split-MNIST using cWGAN. The **first** one studies the percentage of backdoored images (images with a trigger) generated by the replayer, when a varying portion of the training images are backdoored. Our goal is to show that such a percentage is much lower for IAB than for a static backdoor, i.e., the triggers of IAB are much less likely to survive the generative learning. A static backdoor refers to a white square on the image's top left corner. To this end, we trained a binary poison detector based on a training set that is backdoored with the IAB network learned from the first task. Another detector was trained analogously for static backdoor. This allows us to measure the percentage of backdoored images from the replayer. The two detectors achieve 97% accuracy, and similar ideas have been used in evaluating generative models such as inception score (Salimans et al., 2016).

As shown in Figure 4, the static backdoor maintains almost the same percentage of backdoored images used for training, while that for IAB grows much more slowly, producing only 40% backdoored images when 90% of the training images are backdoored. This shows that IAB is far more likely to be discarded by the replayer.

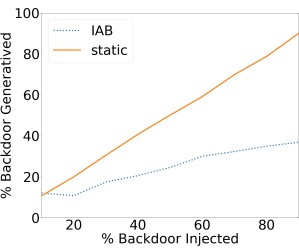 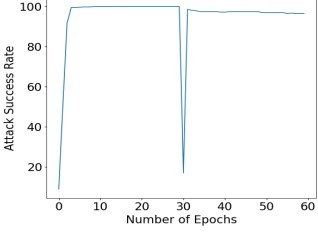 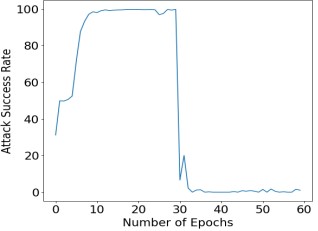

Figure 4: Poison rate of replayed images

(a) Static backdoor      (b) Input-aware backdoor (IAB)

Figure 5: ASR for static backdoor and IAB

Our **second** experiment examines the chance of replaying a backdoored image by using the attack success rate (ASR). In the same setting as the above experiment, we backdoored $25\%$ images in the first 30 epochs with flipped labels, and used them to train a replayer. Then we generated examples from it in the second 30 epochs, and used them to incrementally train a new classifier. This classifier is tested on backdoored images, and the proportion of misclassified images is calculated as the ASR.

If the triggers are preserved by the replayer, then a classifier learned from the replayed images should permit a high ASR. This is confirmed in Figure 5a where the ASR remains at 100% for static backdoor. The drop in the middle is because the new classifier was trained from scratch. In contrast, the ASR drops to zero for IAB in Figure 5b, confirming that the newly trained *classifier* is not backdoored, i.e., the replayed images do not preserve triggers sufficiently well for training a backdoored classifier.

### 5.3 ASSESSING CIAP ATTACK UNDER DEFENSE (OBJECTIVE O3)

We next study how well our attack withstands defenses. To this end, a $\nu$-SVM with a radial basis kernel was applied to the output of the convolutional layer of SpinalVGG. This allows a portion of replayed samples to be filtered out, and the proportion is controlled by $\nu \in (0, 1)$.

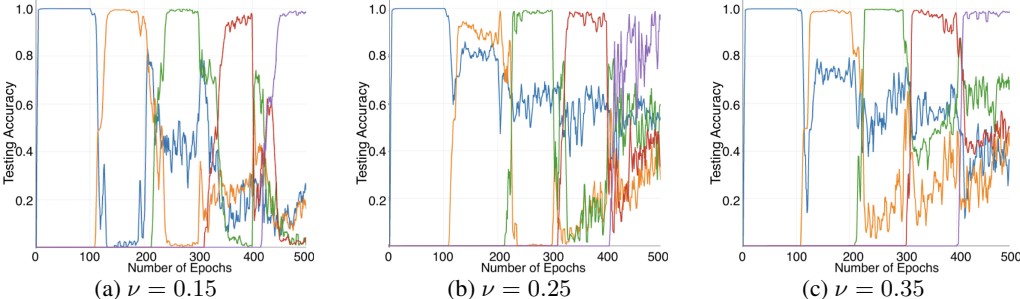

(a) $\nu = 0.15$      (b) $\nu = 0.25$      (c) $\nu = 0.35$

Figure 6: Test accuracy after defense with different $\nu$ values for **split-MNIST**

As shown in Figure 6 where $\nu$ is varied in $\{0.15, 0.25, 0.35\}$ on the split-MNIST dataset, the filtering by $\nu$-SVM does help a little, especially when $\nu$ is set around the poison ratio ($\rho_b = 0.25$). However, it remains unable to well remove the impact of the attack, and the test accuracy on past tasks still falls below 50%. Similar results on the other datasets are relegated to Appendix D.

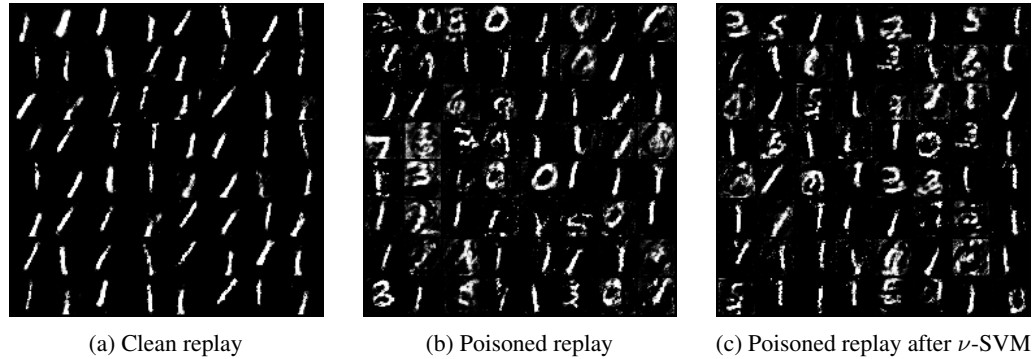

(a) Clean replay      (b) Poisoned replay      (c) Poisoned replay after $\nu$-SVM

Figure 7: Replayed images on **split-MNIST** with label "1" from (a) clean replayer, (b) poisoned replayer, and (c) poisoned replayer after filtered with $\nu$-SVM ($\nu = 0.25$).

The limited improvement could be partially ascribed to the compromised image quality due to the backdoors. To better visualize the consequence of poisoning on the replayer, we compare in Figure 7 the replayed images before and after the attack. Figure 7a presents example images generated by a replayer that is trained on clean images only. In contrast, Figure 7b shows that the poisoned replayer can often generate images from an incorrect class, i.e., another class that is incorrectly labeled as "1". Although the replayer cannot reproduce the input-aware backdoor, it tends to turn the backdoors into some random noise, making it harder for the filter to identify those poisons. As a result, the remaining replayed images are only slightly improved by the $\nu$-SVM filtering as shown in Figure 7c.

We finally investigated the proportion of mislabeled example pairs generated by the replayer. The resulting confusion matrix is shown in Table 1 in Appendix F, using the replayers after completing task 5 on split-MNIST. The total "wrong pair ratio" turns out not high.

## 6 CONCLUSION AND FUTURE WORK

We proposed a novel poisoning attack on the generative replayer in continual learning, so that catastrophic forgetting can be promoted while the accuracy at the current task is not hurt. Our approach takes advantage of input-aware backdoor attacks, whose triggers cannot be well captured by normal generative models thanks to their input dependency. In future work, we will delve more into the theoretical analysis of the trigger-discarding property. We will also extend the approach to continual learning without known task boundaries.

## REPRODUCIBILITY STATEMENT

Anonymous downloadable source code for reproducing experiment results is available at Online Supplementary. Please refer to the included *readme.md* file for details.

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

## A    RESULTS OF FASHIONMNIST-MNIST ON CWGAN

There are two tasks in this dataset: FashionMNIST followed by MNIST. Both tasks have 10 classes, and the victim model's goal is to classify images from them into a *shared* set of labels – "T-shirt" and "hand written 0" are both associated with "class 0". The victim model was trained for 10 epochs with samples from the first task, followed by 10 epochs on the second task.

Figure 8a shows the baseline result without a replayer, where the blue and orange lines represent the test accuracy of the first and second tasks, respectively. As expected, the test accuracy for FashionMNIST declines rapidly from 85% to 48% as the training on MNIST progresses to 10 epochs.

Figure 8b shows the result of DGR-facilitated training. Although the test accuracy of FashionMNIST still drops a bit after MNIST starts to be trained on, it stabilizes at 79%, which illustrates the effectiveness of DGR and the sufficient capacity of the cWGAN.

Figure 8c shows the result after our attack is introduced. In the first task (epochs 0-10), the test accuracy remains very similar to that of Figure 8a and 8b, corroborating the achievement of objective **O2**. When the continual learning moves on to the second task, the test accuracy on the earlier task falls significantly to 61%, much lower than the 79% achieved at the end of the first task. This confirms that the objective **O1** (forgetting) has also been attained.

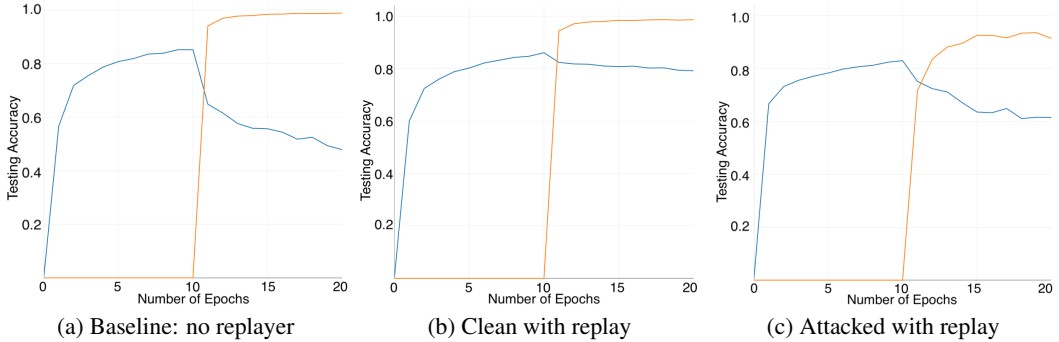

(a) Baseline: no replayer       (b) Clean with replay       (c) Attacked with replay

Figure 8: Test accuracy on clean testing images for **FashionMNIST-MNIST**

## B    RESULTS OF PERMUTED-MNIST ON CWGAN

In this dataset, each task consists of images from all the 10 classes of MNIST. However, each task also employs a unique pixel-level permutation, applied to all the images. The performance is similar to split-MNIST. In Figure 9a where no replayer is used, the test accuracy drops to 20% after new tasks start. Since all the tasks in permuted-MNIST share the same label space, even random guessing would give 10% accuracy. So this 20% is already very close to complete forgetting. Replay ameliorated the problem, but the gain is much obliterated by the CIAP attack in Figure 9c.

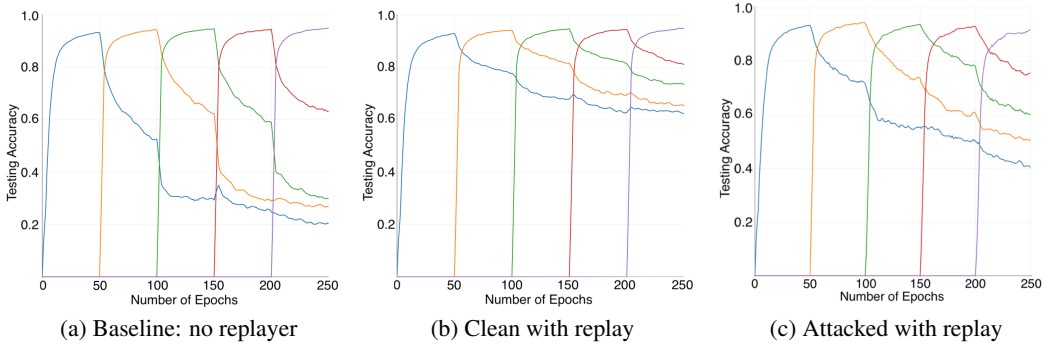

(a) Baseline: no replayer       (b) Clean with replay       (c) Attacked with replay

Figure 9: Test accuracy on clean test images for **permuted-MNIST**

## C    RESULTS OF SPLIT-EMNIST ON CWGAN

Our experiment setup largely follows Shin et al. (2017). Although many continual learning literature also evaluates on 5 tasks or fewer, it is of interest to study more tasks. Note that there are only 10 classes in the split-MNIST dataset, and we have already used up all the classes in the five tasks. Although additional tasks can be constructed by grouping new pairs of classes, it will not serve our investigation well because the sixth task would then include two classes that have already been witnessed before. Although our attacker has promoted catastrophic forgetting, it is still much easier than starting from scratch like in tasks 1-5, i.e., exhibiting no struggle.

As such, we developed a new experiment with the EMNIST dataset (balanced split), which consists of handwritten digits and letters in 47 classes. Since some classes looks alike (i.e. "C" and "c", "S" and "s"), Cohen et al. (2017) merged those similar classes, and balanced the merged classes. We constructed 10 disjoint binary tasks out of it, and the test accuracy on clean test images is shown in Figure 10.

Clearly, the replayer is helpful in alleviating catastrophic forgetting (Figure 10b), and our attacker is able to exacerbate forgetting in all tasks by reducing the test accuracy (Figure 10b). In particular, with the poisoned replayers, all tasks achieve very high accuracy during its own training (Figure 10b). It is noteworthy that in this dataset the help of replayer is not sufficient enough to completely eliminate forgetting. This, however, does not diminish our contribution as an attacker, because the attacker only needs to further impair the accuracy on top of the performance achieved by the replayer.

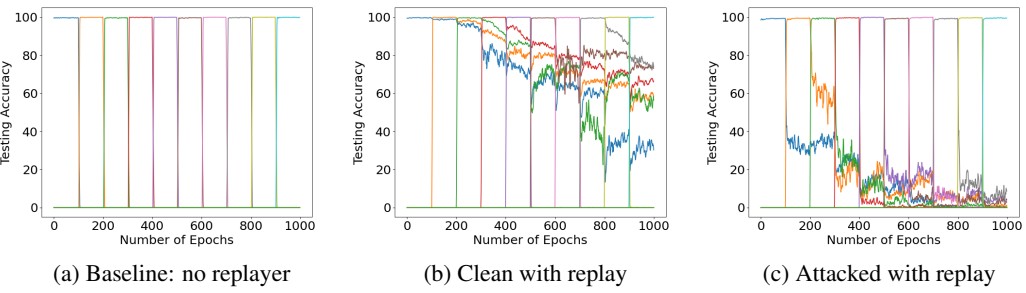

(a) Baseline: no replayer        (b) Clean with replay        (c) Attacked with replay

Figure 10: Test accuracy on clean test images for **split-EMNIST**

## D    RESULTS OF $\nu$-SVM DEFENSE ON SPLIT-CIFAR-10

Similar to the experiment on split-MNIST in Section 5.3, we tested the attack with $\nu$-SVM defense on CIFAR-10. As shown in Figure 11, with three different values of $\nu$, the test accuracy on past tasks dropped to almost 0 after switching to a new task. This confirms that our CIAP remains effective under the defense of $\nu$-SVM.

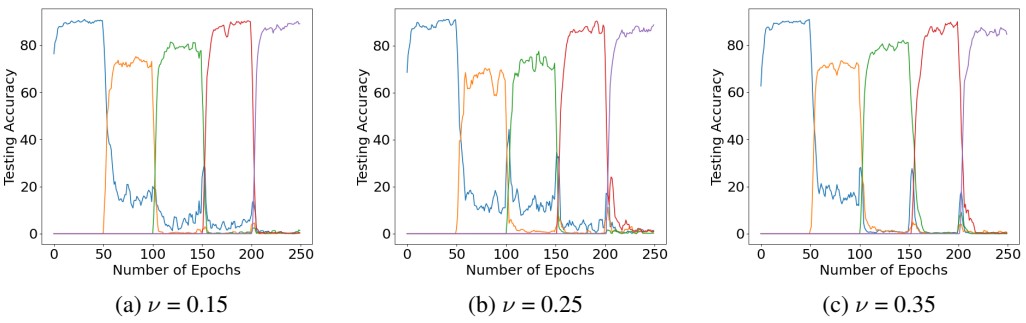

(a) $\nu = 0.15$        (b) $\nu = 0.25$        (c) $\nu = 0.35$

Figure 11: Test accuracy after defense with different $\nu$ values for **split-CIFAR-10**

## E    RESULTS OF FASHIONMNIST-MNIST ON CVAE

Similar to the experiment with cWGAN in Appendix A, we tested the attack with cVAE replayer on FashionMNIST-MNIST. As shown in Figure 12, the testing accuracy on the earlier task dropped from 82% to 24% after attack. This indicates that cVAE is also vulnerable to the proposed attack.

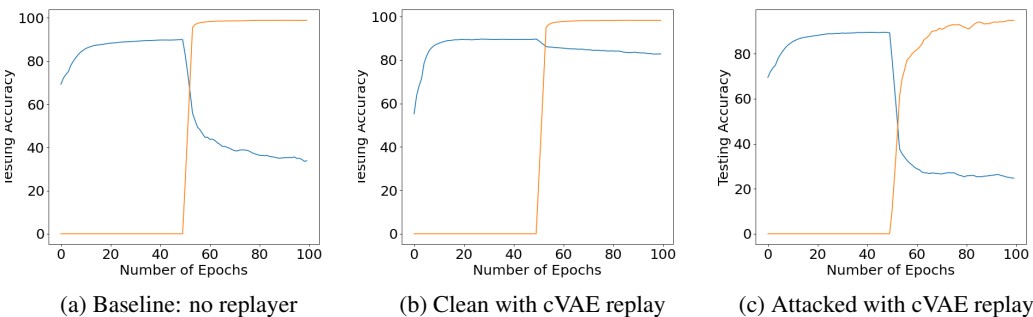

(a) Baseline: no replayer  (b) Clean with cVAE replay  (c) Attacked with cVAE replay

Figure 12: Test accuracy on clean test images for **FashionMNIST-MNIST** based on cVAE

## F    PROPORTION OF MISLABELED PAIRS IN REPLAYED DATA

Table 1: Proportion of mislabeled pairs in replayed data

|   | 0 | 1 | 2 | 3 | 4 | 5 | 6 | 7 | 8 | 9 |
|---|---|---|---|---|---|---|---|---|---|---|
| 0 | **6.25** | 0.15 | 0.89 | 0.18 | 0.61 | 0.23 | 0.7 | 0.09 | 0.21 | 0.63 |
| 1 | 0.01 | **8.19** | 0.31 | 0.02 | 0.1 | 0.08 | 0.01 | 0.49 | 0.63 | 0.1 |
| 2 | 0.03 | 0.14 | **6.26** | 0.98 | 0.07 | 0.67 | 0.39 | 0.42 | 0.62 | 0.36 |
| 3 | 0.01 | 0.08 | 0.65 | **7.45** | 0.74 | 0.1 | 0.18 | 0.26 | 0.12 | 0.35 |
| 4 | 0.02 | 0.09 | 0.05 | 0.04 | **7.49** | 0.46 | 0.22 | 0.53 | 0.44 | 0.62 |
| 5 | 0.03 | 0.04 | 0.03 | 0.19 | 0.48 | **7.09** | 1.21 | 0.04 | 0.46 | 0.39 |
| 6 | 0.1 | 0.11 | 0.07 | 0.04 | 0.04 | 0.23 | **8.92** | 0.11 | 0.06 | 0.27 |
| 7 | 0.13 | 0.36 | 0.2 | 0.09 | 0.13 | 0.16 | 0.02 | **7.42** | 0.21 | 1.22 |
| 8 | 0.01 | 0.01 | 0.0 | 0.0 | 0.3 | 0.0 | 0.04 | 0.01 | **9.08** | 0.49 |
| 9 | 0.0 | 0.0 | 0.0 | 0.04 | 0.15 | 0.11 | 0.01 | 0.01 | 1.62 | **8.02** |

Since the poisoned generator itself does not provide a flag indicating whether a generated feature-label pair is wrong, we trained a classifier $C^*$ on clean MNIST data and applied it to the generated data pairs. The resulting confusion matrix is shown in Table 1 in percentage, using the replayers after completing task 5 on split-MNIST. The columns are the labels produced by $C^*$, while the rows are the labels used to invoke the generator, i.e., used as the label for the replayed data. The total "wrong pair ratio" is 24% (sum of off-diagonal values), which is not that high.

## G  DIFFERENT POISON RATIOS ON SPLIT-MNIST

Similar to Section 5.1, we set up experiments on Split-MNIST with different poison ratios. Please note that the experiment in Figure 13b and Figure 13c were deployed with the attacker not knowing the true label.

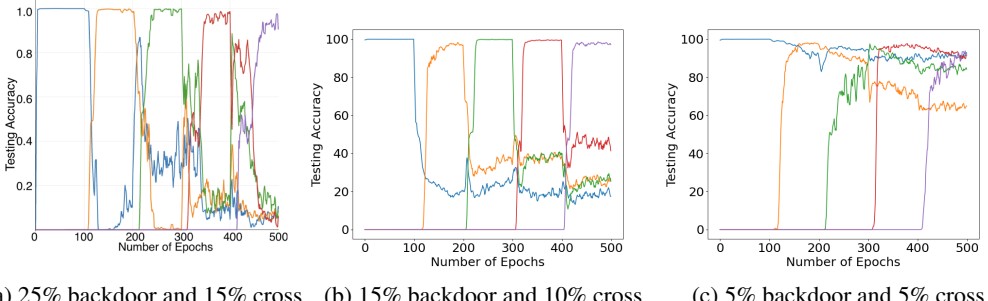

(a) 25% backdoor and 15% cross    (b) 15% backdoor and 10% cross    (c) 5% backdoor and 5% cross

Figure 13: Test accuracy on clean test images for **Split-MNIST**

