# OpenReview forum: "Poisoning Generative Models to Promote Catastrophic Forgetting"
_ICLR.cc/2023/Conference — Submitted to ICLR 2023_

### Official Review · Reviewer_U8Zv · 2022-10-18

**Confidence:** 4
**Clarity, Quality, Novelty And Reproducibility:** good
**Correctness:** 3
**Technical Novelty And Significance:** 3
**Empirical Novelty And Significance:** 3
**Recommendation:** 6

**Strength And Weaknesses:**

Strengths:

- Trendy topic
- Interesting idea

Weaknesses:

- Writing needs to improve
- More experiments are needed


Comments for the authors:

In this paper, the authors exploit the intriguing property of generative models, i.e., they cannot well capture input-aware triggers, to promote catastrophic forgetting, while retaining high accuracy on the current task and evading the defenders to some extent.
The idea of leveraging input-aware triggers in this specific scenario is intriguing, as they are more likely to be overlooked and disregarded by the DGR. Thus, the replayed examples no longer carry the triggers, but they still carry the flipped labels.
More interestingly, the replayed examples cannot be easily filtered out by v-SVM as label-flipped poisons because the input-dependent triggers introduce additional complications to the DGR.

However, I do have the following concerns.

- The paper is a bit hard to follow, so I would suggest the authors re-organize the paper structure. Moreover, as the poisoning attack in the context of continual learning is more complex than common scenarios, it would be much better to provide more visual explanations about the attack process.

- In Section 4, the authors mention they use both Neural Cleansing and v-SVM as defenses. However, only experiments related to v-SVM appear in the evaluation. It would be better to conduct some experiments about Neural Cleansing.

- The attack performance on the FashionMNIST and permuted-MNIST is far less significant than on the MNIST dataset. Thus, I would like to know what will happen if the authors conduct experiments on more complex datasets, e.g., CIFAR100 and CelebA.

Minor:

- In Section 2, "The DRG model" -> "The DGR model"


**Summary Of The Paper:**

This paper proposes a customized poisoning attack in the context of continual learning to stealthily promote catastrophic forgetting.


**Summary Of The Review:**

see strength and weakness

---

> ### Author Response · Authors · 2022-11-18
> **Rebuttal**
>
> **Q1:** Re-organize the paper structure and provide more visual explanations about the attack process.
>
> **A:**  We appreciate the comment.  Could you be more specific about "re-organize" the structure?  And we have provided visualization of poisoned imaged in Figure 2.  So could you be more specific about ``visualizing the attack process''?  Do you mean some diagrams?
>
> **Q2:**  In Section 4, the authors mention they use both Neural Cleansing and $\nu$-SVM as defenses. However, only experiments related to $\nu$-SVM appear in the evaluation. It would be better to conduct some experiments about Neural Cleansing.
>
>
> **A:**
>     Neural Cleansing works by reverse engineering a universal trigger pattern that can turn every image into one specific label.
>     Thus, it will not work when the attack enforced the non-reusability by having "cross" poison.
>     The IAB paper (Nguyen \& Tran, 2020) provides more details and experiments.
>
> **Q3:**
>     The attack performance on the FashionMNIST and permuted-MNIST is far less significant than on the MNIST dataset
>
>
> **A:** We believe this is because for these two datasets, the number of epochs in each task (10 and 50, respectively) are much smaller than that for MNIST (100).  The reason we used only 50 epochs for permuted-MNIST is that, even without applying our attack, the replayer is already not as effective in combating catastrophic forgetting as it did on MNIST.  As a result, the performance would drop too much if we still use 100 epochs.  If we stuck with 100 epochs as in MNIST, our attack (subplot c) would have pushed down the accuracy even more (compared with subplot b).

---

### Official Review · Reviewer_rvck · 2022-10-23

**Confidence:** 4
**Correctness:** 4
**Technical Novelty And Significance:** 2
**Empirical Novelty And Significance:** 2
**Recommendation:** 5

**Clarity, Quality, Novelty And Reproducibility:**

Concerning Related Work:

What is the relationship of this work to previous attacks in Umer et al. 2020, "Targeted Forgetting and False Memory Formation in Continual Learners through Adversarial Backdoor Attacks" and follow-up work in Umer 2021 and especially Umer 2022 "False Memory Formation in Continual Learners Through Imperceptible Backdoor Trigger"? It would be great if the submission could discuss and distinguish the proposed attack from related work on poisoning attacks in continual learning. Somewhat related is also Li and Ditzler "Targeted Data Poisoning Attacks Against Continual Learning Neural Networks".

Concerning the Title:

I believe the current title is not a great fit for this work. While generative models are indeed poisoned to promote catastrophic forgetting, this is not a generic attack against generative models, but an attack that happens only in the context of generative models used as replay systems in continual learning. I specifically do not think it a bad thing that the attack targets this threat model, just that this is not well reflected in the title. A suggestion could be "Poisoning Generative Replay in Continual Learning to Promote Catastrophic Forgetting".

Concerning Clarity of Writing:

The submission is overall well-motivated and introduced concepts are well-explained. However, the presentation is held back a bit by a smaller stream of typos and grammatical errors. For example, on the first pages:
* "...urges investigation of poisoning attacks on it in a broader range of learning paradigms"->"...urges the investigation of poisoning attacks in a broader range of learning paradigms"
* " and work in batch." -> "and work in batched learning scenarios"
* "a novel customization of dirty-label input-aware backdoor "->"a novel customization of dirty-label input-aware backdoors"
* "The poison should sustain solid defense deployed by"->"The poisoned data should be robust to solid defenses deployed by"
* " the transiency of data stream" -> " the transiency of data streams"
* "The DRG models can be simplified" -> "DGR models can be simplified"
*  "we are now missioned to" -> "We are now tasked to "

My review is not changed by these problems with the writing, but I do think this work would be more impactful if its another pass over the text would be made with particular focus on the writing, especially in earlier sections.


**Strength And Weaknesses:**

The proposed attack is interesting and overall executed well, and the authors spend a good amount of effort on motivating and deriving it in detail. From the experiments, I particularly like analysis in sections 5.2 and inclusion of baseline defenses in 5.3. The experimental section is somewhat limited by the small scale of experiments investigated here, it would have been interesting to see whether this attack is also possible if the dataset (and by extension the generator) have to model more complex behavior, e.g. when looking at variations of ImageNet for continual learning.

I've listed some weaknesses of the submission in its current state below:

* The submission contains motivation and discussion of the cross-trigger examples, but I could not find an ablation detailing the effectiveness of this modification and effectiveness in relationship to cross ratio described in the experimental evaluation.

* Further, what about baseline attacks? A simple one that comes to mind is simply maliciously flipping the labels of 40% of the data, e.g. by  predictably flipping between the classes in each task.

* I wonder whether the authors can comment on the necessity of dirty-label backdoors in this system. Is the proposed attack contingent on having access to label information on poisoned samples? It would be great if this point could be clarified somewhere in the submission.

* Finally, if I understand it correctly, the attacks described in Sec. all assume that 25%+15% of the data can be poisoned. What is the attack performance as these numbers are increased or decreased? It would be great if the whole range could be plotted.

**Summary Of The Paper:**

This submission presents a data poisoning attack against continual learning. Specifically, continual learning systems that utilize generative algorithms to remember and replay data from previous tasks, are attacked in this work. The attack inserts poisoned data (containing backdoored data and flipped labels) that later breaks the generator when it replays samples.

**Summary Of The Review:**

Overall I like the idea described in this submission, but some open questions concerning baselines, ablations and related work remain for me, as outlined above. I would like for the authors to comment on and clarify these points before recommending acceptance of this submission.

---

> ### Author Response · Authors · 2022-11-18
> **Rebuttal**
>
> **Q1:** The submission contains motivation and discussion of the cross-trigger examples, but I could not find an ablation detailing the effectiveness of this modification and effectiveness in relationship to cross ratio described in the experimental evaluation.
>
>
> **A:** The cross-trigger is to enforce the non-reusability of triggers,
> which helps our attacker to withstand the Neural Cleanse defense.
> It is discussed in IAB paper.
>
> **Q2:** what about baseline attacks? A simple one that comes to mind is simply maliciously flipping the labels of 40\% of the data, e.g. by predictably flipping between the classes in each task.
>
> **A:**
> Adding random images with random labels will not work.
> In continual learning, there is a trained classifier (or classifiers) at all tasks, except the first task.
> So the mis-labeled images will be easily detected by the classifier,
> hence failing objective O2.
> In contrast, the backdoor will make sure that the trained classifier still classifies the triggered images into the desired (wrong) class,
> hence being less detectable.
> This is why backdoor training examples play a key role in our attacking method.
> We have lightly touched this point in the third line of the "Our contribution" paragraph on page 2, but we will elaborate on it in the revision.
>
>
> **Q3:** I wonder whether the authors can comment on the necessity of dirty-label backdoors in this system. Is the proposed attack contingent on having access to label information on poisoned samples? It would be great if this point could be clarified somewhere in the submission.
>
>
> **A:**
>     We indeed do not require knowing the true label.
>     However, we do need to use dirty labels to perform the attack.
>     In the experiment for the next question (Q4),
>     we have dropped the access to the true label and just randomly selected a label as the target.
>     There is a small chance that such a label is indeed correct.
>     But given the large number of classes, this is quite unlikely in practice.
>
> **Q4:** Finally, if I understand it correctly, the attacks described in Sec. all assume that 25\%+15\% of the data can be poisoned. What is the attack performance as these numbers are increased or decreased? It would be great if the whole range could be plotted.
>
>
> **A:**     This is an interesting investigation.
>     We added experiments with lower poison ratios in Appendix G.
>     The result shows that with 15\%+10\% (dirty label backdoor examples + cross examples),
>     the attack brings clean test accuracy on past tasks down to ~30\% (Figure 13b).
>     However, after further limiting the ratio to 5\%+5\%,
>     the clean test accuracy remains above 80\% (Figure 13c).
>     As 25\% poison is already high,
>     we did not test the effect of further increasing it.
>
> **Q5:** New title and typos
>
> **A:** We have fixed the typos in the revision.
> We are grateful for the proposed title, and we will adopt it.
> As Openreview does not allow changing the title and it will cause confusion if it is only changed in the PDF,
> we defer the change to the future.

---

> > ### Comment · Reviewer_rvck · 2022-11-27
> > **Response**
> >
> > Thank you for clarifying these points. I have no further questions aside from one regarding Q2. You mention that a baseline attack would reduce validation accuracy within the current task, but objective O2 is a bit vague on what would constitute a significant drop in accuracy. A model with 40% of labels flipped might still have a very decent baseline validation accuracy. Aside, such a baseline would be interesting as a point of reference, even if it violates one of the assumptions of stealth.

---

> > > ### Author Response · Authors · 2022-12-05
> > > **Q2**
> > >
> > > Thank you for the follow-up. We conducted the recommended baseline attack on Split-MNIST by flipping 25% labels without changing the corresponding images. Since Openreview no longer accepts PDF revisions, we uploaded the updated paper to the online supplementary Dropbox folder, whose URL was provided at the time of submission:
> > >
> > > https://www.dropbox.com/sh/mku8oln1t7ngscl/AABVPSwZBlx41GtQYRyYVRgha?dl=0
> > >
> > > The file name is Paper_Rebuttal_update.pdf.  As shown in Figure 14b of Appendix H, starting from the second task (orange line), the test accuracy of current tasks drops to below 20\%, compared with 100\% without attack (Figure 14a).  Thus, the baseline attack can be easily discerned by the victim during training of the current task, violating the objective O2.

---

### Official Review · Reviewer_YjaT · 2022-10-27

**Confidence:** 4
**Clarity, Quality, Novelty And Reproducibility:** The paper is clear to read.
**Correctness:** 2
**Technical Novelty And Significance:** 2
**Empirical Novelty And Significance:** 2
**Recommendation:** 3

**Strength And Weaknesses:**

I found that the problem setting is very narrow and has limited applicability.  My major concerns are as follows:

1. It seems like one can easily defend the proposed attack simply by not solely relying on a generative replayer. Instead, all you need is to keep a small dataset for each task in addition to the generative model, and the attack scenario (and the proposed algorithm) becomes invalid. More specifically, a small dataset for each task can be used to check the validity of the generated data.

2. The proposed approach does not have anything to do with "promoting forgetting." What it does is inject wrongly labeled inputs that cannot be generated by a generative replayer. It's unclear to me why one needs to backdoor training samples to achieve their goal. Indeed, arbitrary random inputs with random labels might be sufficient to achieve the same goal. Note that both attacks are easy to detect if someone inspects the generated data and compare it with labels before using it for training.

3. Experimental results are highly limited. It will be great if the authors can use standard benchmark datasets for continual learning.

It would be great if the authors could correct my misunderstanding if any.

**Summary Of The Paper:**

The authors propose a poisoning attack on the generative model that is used to generate past data in the continual setting.

**Summary Of The Review:**

The problem setting is not clearly motivated, and the proposed solution's validity is unclear. See my major concerns above.

---

> ### Author Response · Authors · 2022-11-18
> **Rebuttal**
>
> **Q1:**  Keeping a subset of data from past tasks.
>
> **A:** One of the basic assumptions in continual learning is that we cannot access data from past tasks.
> Besides space complexity,
> privacy concerns may forbid storing any data,
> especially in clinical applications [3].
> In the case that storing past data is permitted,
> catastrophic forgetting can be directly addressed by it,
> eliminating the need of replayer in the first place.
>
>
> **Q2:** The proposed approach does not have anything to do with "promoting forgetting." What it does is inject wrongly labeled inputs that cannot be generated by a generative replayer. It's unclear to me why one needs to backdoor training samples to achieve their goal. Indeed, arbitrary random inputs with random labels might be sufficient to achieve the same goal. Note that both attacks are easy to detect if someone inspects the generated data and compare it with labels before using it for training.
>
>
> **A:** We believe this is grave misunderstanding.
> Adding random images with random labels will not work.
> In continual learning, there is a trained classifier (or classifiers) at all tasks, except the first task.
> So the mis-labeled images will be easily detected by the classifier,
> hence failing objective O2.
> In contrast, the backdoor will make sure that the trained classifier still classifies the triggered images into the desired (wrong) class,
> hence being less detectable.
> This is why backdoor training examples play a key role in our attacking method.
> We have lightly touched this point in the third line of the "Our contribution" paragraph on page 2, but we will elaborate on it in the revision.
>
>
> All dirty label data poisoning attack can be easily detected by human inspectors, but it is generally not feasible to hire human inspectors to check all training data.
> As for automated inspectors,
> we have discussed several of such defense methods in Section 5.3,
> and showed that the proposed CIAP attack is still effective in their presence.
>
> **Q3:** More benchmark datasets for continual learning
>
> **A:**  We have conducted experiments in split-MNIST and permuted-MNIST, which are also used in DGR paper.
> In addition,
> we experimented dataset with more tasks: split-EMNIST,
> dataset in colored space: CIFAR-10,
> and domain incremental dataset: FashionMNIST-MNIST.
> If we consider continual learning papers **with attacks**,
> we have included all "standard datasets".
> For example, [1] used permuted-MNSIT and split-MNIST,
> and [2] used MNIST-FashionMNIST-KMNIST and rotated-MNIST.
> We are actually having more challenging datasets covered in experiments such as CIFAR10.
>
> [1] Umer et al. 2020, "Targeted Forgetting and False Memory Formation in Continual Learners through Adversarial Backdoor Attacks",
>
> [2] Li and Ditzler "Targeted Data Poisoning Attacks Against Continual Learning Neural Networks" were all using those datasets.
>
> [3] Lee CS, Lee AY. Clinical applications of continual learning machine learning. Lancet Digit Health. 2020 Jun;2(6):e279-e281. doi: 10.1016/S2589-7500(20)30102-3. PMID: 33328120; PMCID: PMC8259323.

---

### Official Review · Reviewer_jjXs · 2022-10-30

**Confidence:** 4
**Correctness:** 2
**Technical Novelty And Significance:** 2
**Empirical Novelty And Significance:** 3
**Recommendation:** 5

**Clarity, Quality, Novelty And Reproducibility:**

The quality of the writing could use improvement. The paper seems somewhat novel.

**Strength And Weaknesses:**

### Strengths:

1. I agree with the authors that the poisoning attacks on continual learning systems are understudied (especially with the overlap with FL here).
2. The high-level idea of somehow poisoning the generative model for continual learning seems quite interesting.

### Weaknesses:
1. The paper is poorly written, and clunky to read. I would highly recommend the authors do a more thorough read-through of the work to make it more easy digest. Note - I added this comment as a weakness, but it did not factor into my recommendation.
2. The threat model is quite loose - plenty of modern poisoning attacks work in the clean-label regime, as this is far more convincing as a threat, and I am wary of any attacks that leverage manipulated labels. Also, the assumption that the attacker can query the victim model's exact gradient seems unrealistic. At least for some of the works that the authors reference as making similar assumptions, these claims are erroneous and in fact no such assumptions are made.
3. It's unclear what exactly are novel contributions of the paper - the poisoning method seems to just reuse IAB... From my understanding, the paper's novel contributions are the application of this to the setting of continual learning, which is interesting, but somewhat minor.
4. It's unclear to me why a trigger is necessary at all? Why not just insert mis-labeled data, with the hope that the generative model produces incorrect data for a certain label during a future replay?
5. There are a lot of assertions that at best have no justification, and often just seem flat out wrong intuitively. For example:
    * "If the trigger is a constant small white square at the image center, most generative models will preserve it"
    *  "Since the mislabeled examples for the current task all carry an input-aware trigger, [...] it predicts accurately on pristine test examples for task t which carry no trigger"
6. The experiments are on very simple datasets.
7. I'm not an expert on DGR, but it would be nice if experiments were done with more commonly used diffusion-based generative models.

**Summary Of The Paper:**

The paper proposes a method to poison the generative models used in DGR in an effort to degrade continual learning methods using DGR.

**Summary Of The Review:**

Because of the strong attack assumptions, and the unsubstantiated claims in the paper, I do not vote for acceptance at this point. However, I think the core idea is quite interesting, and very much encourage the authors to correct the writing errors, address reviewer concerns, and resubmit in the future.

---

> ### Author Response · Authors · 2022-11-18
> **Rebuttal (Part I)**
>
> **Q1:** Writing
>
> **A:** Can you be more specific about which part or aspect of the paper is hard to read?  Reviewer YjaT comments that \``The paper is clear to read.'', and Reviewer rvck says \``The submission is overall well-motivated and introduced concepts are well-explained.''  Reviewer U8Zv says \``good'' on ``Clarity, Quality, Novelty And Reproducibility''.
>
>
> **Q2:** Assumption of access to gradient
>
> **A:** The ``Dynamic backdoors (Salem et al., 2020b)" we cited was regretfully a typo.  We actually intended to cite the IAB (Nguyen \& Tran, 2020), which requires access to gradients.  The other two---Witches’ brew in Geiping et al., (2021) and implicit differentiation based poisoning (Muñoz-González et al., 2017)---remain correctly cited, and they do use the gradient.
>
> In evasion attack, projected gradient ascent is commonly used and surely they require the gradient.
> Gradually, the area progressed to drop this requirement by, e.g., using a blackbox attacker.
> Since poisoning generative replay in continual learning is a new and very challenging task, we think it reasonable to start by assuming access to the gradient.
>
>
> **Q3:** It's unclear what exactly are novel contributions of the paper
>
> **A:**  We respectfully disagree with this assessment. We have summarized our novelty in the three bullets in the introduction.  The machine learning community tends to underrate the novelty of combining two existing methods to achieve an unprecedented goal. Although this assessment often makes sense, it still needs to be scrutinized case by case, with careful immunity to the post-hoc bias of ``everything is easy, once you know how" (or be informed of how). A good criterion we use when reviewing such papers is to ask ourselves whether we can find a solution to this problem by ourselves, provided that the problem is significant and important (which is recognized by all the reviewers of our paper).
>
> We believe our use of input-aware backdoor attack to achieve objectives O1 to O3 in continual poisoning is clearly innovative and highly nontrivial. For example, identifying and leveraging the trigger-discarding properties of generative models, and using backdoor attacks to achieve poisoning attacks in continual learning. Similarly, no new algorithm was proposed in the seminal paper of "Intriguing Properties of Neural Networks" (just projected gradient ascent), but it is the sharp insight that really counts. We would like to invite reviewers who do not quite recognize our technical innovation to propose an alternative solution. We will be happy to continue the discussion, which in our experience, will most effectively illuminate how much our solution is meticulously crafted and innovative.
>
>
> **Q4:**  Why not just insert mis-labeled data, with the hope that the generative model produces incorrect data for a certain label during a future replay
>
> **A:** This proposal will not work.  In continual learning, there is a trained classifier (or classifiers) at all tasks, except the first task.
> So the mis-labeled data will be easily detected by the classifier,
> hence failing objective O2.
> In contrast, the trigger will make sure that the trained classifier still classifies the instance into the desired class, hence being more stealthy.
> We have lightly touched this point in the third line of the "Our contribution" paragraph on page 2, but we will elaborate on it in the revision.
>
> In contrast, replayed images cannot be detected by these classifiers.
> This has been detailed in the last paragraph of Section 4, which we replicate here:
>
> ``It is crucial to recognize that the replayed examples are not simply label-flipped poisons (i.e.,
> clean images with a wrong label), although the replayer is poisoned with label-flipped and backdoored examples. This is for two reasons. Firstly, since the examples of a class $y'$ is fed to the replayer to train for the class $y$, the generation of the features/images for class $y$ is contaminated. Secondly, the input-dependent triggers introduce additional complications to the generative model. Indeed, we tested by directly generating label-flipped examples based on clean images, and $\nu$-SVM easily filtered them out. However, this is not the case when $\nu$-SVM is applied to our replayed images (\S 5.3).''
>
> **Q5.1:** Assumption ``If the trigger is a constant small white square at the image center, most generative models will preserve it".
>
> **A:** This phenomenon is demonstrated in Section 5.2, where we shown that the static backdoors are preserved while the dynamic ones are dropped.
>
> **Q5.2.** Assumption``Since the mislabeled examples for the current task all carry an input-aware trigger, [...] it predicts accurately on pristine test examples for task $t$ which carry no trigger".
>
> **A:** This phenomenon is demonstrated in Section 5.1.  For example, in Figure 1(c), the first task (blue line) has nearly 100\% accuracy with clean testing inputs on the first 100 epochs.

---

> > ### Author Response · Authors · 2022-11-18
> > **Rebuttal (Part II)**
> >
> > **Q6:** Datasets
> >
> > **A:** The datasets in our paper are all standard in continual learning with attacks.
> >     We have conducted experiments in split-MNIST and permuted-MNIST, which are also used in DGR.
> > In addition, we experimented dataset with more tasks: split-EMNIST, and colored dataset: CIFAR-10, and domain incremental dataset: FashionMNIST-MNIST.
> > If we consider continual learning papers **with attacks**, we have included all "standard datasets". For example, [1] used permuted-MNSIT and split-MNIST, and [2] used MNIST-FashionMNIST-KMNIST, and rotated-MNIST.
> > We are actually having more challenging datasets covered in experiments such as CIFAR10.
> >
> > **Q7:** Other generative models
> >
> > **A:**
> >     DGR is one of the most effective models for continual learning.  Besides the GAN-based model, we also added experiments with VAE for the replayer. Although diffusion-based models can be more effective as a replayer, they are more sophisticated than cWGAN and cVAE, hence more susceptible to attacks.  Since cWGAN and cVAE are already performing well on the continual learning tasks in our experiments, we leave the test on diffusion models to future work.
> >
> >
> > [1] Umer et al. 2020, "Targeted Forgetting and False Memory Formation in Continual Learners through Adversarial Backdoor Attacks",
> >
> > [2] Li and Ditzler "Targeted Data Poisoning Attacks Against Continual Learning Neural Networks" were all using those datasets.

---

> > ### Comment · Reviewer_jjXs · 2022-11-22
> > **Response**
> >
> > > Can you be more specific about which part or aspect of the paper is hard to read
> >
> > Sure. I recognize my initial comment was not very constructive as it didn't point to specific parts, so I apologize for that. After another pass, here are a few paragraphs/sentences that are either clunky to read, or contain wording/grammatical errors:
> >
> > * "Secondly, the transiency [...] start to occur."
> > * "Our contribution, therefore, [...] soon in the future"
> > * The paragraph beginning with "DRG models can be simplified [...]"
> > * "In backdoor attacks, [...] clean test examples"
> > * "Such a flexibility of modulation proves essential."
> > * The first paragraph in 3.3
> >
> > Additionally, I don't think the use of O1, ... and P1, ... is helpful for reading the paper as the reader either has to memorize these abbreviations or scroll back every time one of these is encountered. I would strongly recommend for the authors to more explicitly refer to these concepts.
> >
> > Finally, I think in the rebuttal revision, there may be an extra page added, with a blank page in the middle.
> >
> > I would like to reiterate, though, that these points did not factor into the score, and could be easily corrected, so I do not hold it against the authors.
> >
> > &nbsp;
> >
> > > The other two---Witches’ brew in Geiping et al., (2021) and implicit differentiation based poisoning (Muñoz-González et al., 2017)---remain correctly cited, and they do use the gradient.
> >
> > Witches' Brew does not have access to any victim gradient. They train a surrogate model to estimate this quantity.
> >
> > &nbsp;
> >
> > > Since poisoning generative replay in continual learning is a new and very challenging task, we think it reasonable to start by assuming access to the gradient.
> >
> > It's not just the access to the gradient that I think is unreasonable - I also think the percentage of data poisoned, and the non clean-label assumptions of this work make the threat model unconvincing. I understand your point about it being a first-attempt at poisoning DGR, but the threat model is so loose I'm not convinced that the attack warrants publication at a top-tier conference.
> >
> > &nbsp;
> >
> > > The machine learning community tends to underrate the novelty of combining two existing methods to achieve an unprecedented goal
> >
> > I'm not sure about making statements about a larger trend, but I am convinced by your argument here. To be clear, I think that the novelty alone is not a "fatal" flaw of the work, but rather, for this reviewer at least, the combination of the minor technical novelty combined with the very loose threat model makes the paper less impressive. I do think that the idea of poisoning DGR is creative, and I applaud the authors for the novelty of the project (just not the technical aspects).
> >
> > &nbsp;
> >
> > > A4, A5.1
> >
> > Thanks for the explanation.
> >
> > &nbsp;
> >
> > > A5.2
> >
> > Thanks for this explanation. I am a bit surprised by the result as it seems like the mislabeled examples (even with the trigger) would induce an accuracy drop on clean test examples, but I accept the authors' explanation here.
> >
> > &nbsp;
> >
> > I have raised my score to a 5 in light of the authors response, however, I still lean toward rejection as the threat model is incredibly loose, and this makes the empirical results less convincing.

---

### Decision · Program_Chairs · 2023-01-20

**Decision:**

Reject

**Justification For Why Not Higher Score:**

Limited applicability of the threat model, attacks being contrived, poor writing, and limited experiments.

**Justification For Why Not Lower Score:**

N/A

**Metareview: Summary, Strengths And Weaknesses:**

Reviewers appreciated that the core idea was somewhat interesting. However, criticisms included limited applicability of the threat model, attacks being contrived, poor writing, and limited experiments. The reviewers felt the drawbacks outweighed the virtues, and unanimously voted against acceptance.